# The Effect of Progressive Resistance Exercise Training on Cardiovascular Risk Factors in People with Intellectual Disabilities: A Study Protocol

**DOI:** 10.3390/ijerph192416438

**Published:** 2022-12-08

**Authors:** Roy G. Elbers, Kirsten I. de Oude, Theodore Kastanidis, Dederieke A. M. Maes-Festen, Alyt Oppewal

**Affiliations:** Department of General Practice, Intellectual Disability Medicine, Erasmus MC, University Medical Center Rotterdam, 3015 GD Rotterdam, The Netherlands

**Keywords:** cardiovascular disease, resistance training, exercise, intellectual disabilities

## Abstract

Progressive resistance exercise training (PRET) reduces cardiovascular risk factors (CVRF) in the general population. It is unknown if PRET also reduces these risk factors in adults with intellectual disabilities (ID). The aim is to present the protocol of an intervention study that investigates the effect of PRET on CVRF in adults with ID. We will use a repeated time series design with one study group. Adults with mild-to-moderate ID and at least two CVRF are eligible (Netherlands Trial Register, NL8382). During a 12-week baseline period, measurements take place at a 6-week interval. After this, the PRET programme starts for 24 weeks, after which all measurements will be repeated. We will use hierarchical regression models, adjusted for sport activity and medication use, to estimate the effect of PRET. After the intervention, the participants will be followed-up for 12 weeks. We will evaluate factors for successful implementation of exercise in daily life. Primary outcomes are: hypertension, obesity, hypercholesterolemia, diabetes, metabolic syndrome. Secondary outcomes are: physical fitness, sarcopenia, physical activity, activities of daily living, falls, challenging behaviour. If our results show that the PRET programme is effective, it may be a promising non-pharmacological intervention to reduce CVRF in adults with ID.

## 1. Introduction

The life expectancy of people with intellectual disabilities (ID) is increasing and approaching that of the general population [1,2]. However, an increased life expectancy does not mean that people with ID are ageing healthy. The Healthy Ageing and Intellectual Disabilities (HA-ID) study found that cardiovascular risk factors (CVRF) were highly prevalent in adults with ID of 50 years and over. Hypertension was seen in 53% of the participants, obesity in 46–48%, hypercholesterolemia in 23%, diabetes in 13%, and metabolic syndrome in 45% [3].

When CVRF are present, it is recommended to use either medication or promotion of a healthy lifestyle to prevent the development of cardiovascular disease [4]. Because adults with ID are more likely to be exposed to polypharmacy [5], promoting a healthy lifestyle and physical activity may have an even more important role in reducing cardiovascular risk. Unfortunately, most adults with ID have an inactive lifestyle with only 9% of them achieving physical activity levels needed for gaining health benefits [6].

Resistance exercise training is a well-studied intervention that reduces CVRF in the general population [7,8,9,10] and may be promising to reduce CVRF in adults with ID as well. For adults with cardiovascular and metabolic disease, the American College of Sports Medicine (ACSM) guidelines recommend a progressive resistance exercise training (PRET) programme at vigorous intensity for all large muscle groups [11]. Although research has been conducted in the general population, it is still unknown whether PRET is beneficial for reducing CVRF in adults with ID. Studies indicate that the physiological responses to exercise are different in people with ID. For example, there are indicators that during exercise they have a blunted blood pressure and baroreflex sensitivity response [12,13], and a lower maximal heart rate and peak oxygen uptake [14] than people in the general population. Therefore, the effect of exercise interventions may differ for this population, and we cannot generalize the results from the general population to people with ID.

Due to low physical activity levels [15], adults with ID are prone to other health problems. Low levels of physical activity can result in loss of muscle strength and muscle mass which can lead to sarcopenia at older age [16]. Sarcopenia is characterised by progressive and generalised loss of skeletal muscle mass and strength with a risk of adverse health outcomes [17]. Sarcopenia has been found prevalent in older people with ID, at already younger ages than in the general population [17,18]. In the general population resistance training reduces sarcopenia [19], as it increases the muscle strength and muscle mass [20]. Other studies showed that people with ID have poor balance, strength, muscular endurance, and gait speed [15,21] which can lead to difficulties in performing activities of daily living and thereby increase the dependency of caregivers [22] and falls. In addition to these health problems, challenging behaviour, such as aggression and self-injury, is common [23]. Team sports seem to reduce challenging behaviour [24], but it is still unknown if resistance training is effective in reducing challenging behaviour in adults with ID.

A PRET programme may enhance physical activity and be beneficial for people’s cardiovascular risk profile and health in general. In a previous study, we composed a Resistance Exercise Set for adults with ID (RESID) [25]. We found that these exercises were feasible to perform by adults with mild to moderate ID and CVRF [26]. Therefore, the RESID can be used in the PRET programme as the content of the training sessions. This programme may be an additional intervention to existing pharmacological treatments for CVRF in adults with ID. To our knowledge, no study investigated the effect of PRET on reducing the above-described health issues in this population. The primary aim of this study is to investigate the effect of the PRET programme on CVRF (hypertension, obesity, hypercholesterolemia, diabetes, and metabolic syndrome) in adults with ID. Second, we will investigate the effect of PRET on physical fitness, sarcopenia, physical activity, activities of daily living, falls, and challenging behaviour. This paper presents the study protocol, outcomes, and statistical analysis plan.

## 2. Material and Methods

### 2.1. Study Design

This study has a repeated time series design with one study group. The study period will last one year and is divided into a 12-week baseline period, a 24-week intervention period (PRET programme), and a 12-week follow-up period. We will use the baseline measurements for control comparisons.

This study follows the principles of the Declaration of Helsinki [27] and is in accordance with the Dutch Medical Research Involving Human Subjects Act. We obtained ethical approval from the Medical Ethical Committee of the Erasmus MC, University Medical Center Rotterdam in the Netherlands (MEC-2019-0544). The study is registered in the Netherlands Trial Register (NTR, NL8382). Written informed consent will be obtained from all participants or their legal representatives.

### 2.2. Study Population, Recruitment, and Inclusion

This study will be conducted within the Healthy Ageing and Intellectual Disability (HA-ID) consortium. The HA-ID consortium is a collaboration between the Chair of Intellectual Disability Medicine of the Erasmus MC, University Medical Center Rotterdam and three care providers in the Netherlands specialised in care for people with ID: Ipse de Bruggen (Zoetermeer), Abrona (Huis ter Heide), and Amarant (Tilburg).

Figure 1 summarises the inclusion and exclusion criteria for this study. Participants must meet the following criteria to be included: (1) being 18 years or older; (2) living in a residential facility or receiving care from one of the participating care providers; (3) diagnosed with mild (IQ = 50–69) or moderate (IQ = 35–49) ID [28]; (4) diagnosed with metabolic syndrome [29], or at least two of the following CVRF: hypertension [30], hypercholesterolemia [4], diabetes mellitus type 2 [31], or obesity [32]. Metabolic syndrome is defined as the presence of any three of the following CVRF: elevated waist circumference, elevated blood pressure, elevated fasting glucose, elevated triglycerides, and low high density lipoprotein cholesterol [29]. Cut-off values for the CVRF are summarised in Table 1. Furthermore, participants must be able to walk independently and able to follow instructions on posture and performance for good execution of the exercises.

We will develop a query to screen medical files for potential participants. Potential participants are screened by a physical therapist for their eligibility and expected adherence to the PRET programme. To identify health issues that make participation in the PRET programme unsafe, the professional caregiver of eligible participants will fill out the Physical Activity Readiness Questionnaire (PAR-Q) [33]. If a positive response to any of the items of the PAR-Q is given, the participant’s physician will do a health check to assure safe participation. Participants who do not obtain medical clearance from their physician or have physical limitations (including neurological and orthopaedic conditions) interfering with participation in the PRET programme will be excluded from the study.

### 2.3. Informed Consent Procedure

All potential participants will receive study information and a consent form. A behavioural scientist will assess if potential participants understand the easy-read study information and if they are able to decide for participation themselves. If potential participants are not able to make an informed decision, we will also send the study information and a second consent form to their legal representative and ask them for a decision.

### 2.4. Intervention

The PRET programme will take 24 weeks and consists of two sessions per week, with at least 48 h between the sessions. The intervention will be led by experienced physiotherapists and physical activity instructors employed by the care providers. All instructors take part in a training prior to the study. In this training, the intervention protocol and exercises will be discussed and practiced ensuring that the intervention is delivered according to the protocol. Every session consists of seven exercises (RESID) to train all large muscle groups [25]. For each muscle group, alternative exercises will be provided for participants who are not able to perform the original exercise with a correct execution. The instructor-participant ratio will be 1 to 1 or 1 to 2.

Table 2 summarises the training parameters of the PRET programme. The programme is in accordance with the ACSM guidelines [11] and characterised by four mesocycles. Each mesocycle consists of six weeks of training. After each mesocycle the training intensity will increase gradually, and within each mesocycle the weights will be adjusted in a way that the intensity remains equal. Training intensity will be measured by the percentage of the heaviest load one can lift for one repetition (% of 1RM) [34]. In the first mesocycle, participants are introduced to the resistance exercises and familiarised to the training protocol. They will learn to perform the exercises safely and with good posture. During mesocycles 2, 3, and 4, the number of repetitions will be gradually decreased while the training intensity will be gradually increased until vigorous intensity is reached in the fourth mesocycle (80% of 1RM). The resting time between sets will depend on the training intensity and varies between 30 s and 2 min.

The instructors will register attendance, training intensity, performed number of sets and repetitions, and the weights used for each exercise in a log. To maximise attendance and motivation, each participant receives a progression certificate to track their progress and a T-shirt to wear during the training and measurements. At completion of the programme, the participant receives a certificate and a medal.

### 2.5. Co-Interventions

Participants will continue their usual activities but cannot start another intervention or activity to improve their physical activity level during the study period. Medical interventions that are necessary for regular care are allowed to start. All changes in sport activities and medication use related to CVRF will be recorded.

### 2.6. Primary Outcomes

The following CVRF are the primary outcomes.

#### 2.6.1. Hypertension, Blood Pressure

A trained researcher will use the Omron-M7 (Omron Healthcare, Kyoto, Japan) to measure blood pressure from both arms after sitting for two minutes without talking. The Omron-M7 has been found valid according to the international protocol of the European Society of Hypertension [35]. We will select the arm with the highest value for the actual measurement. This arm will be measured twice, and the average score (mm/Hg) for systolic (SBP) and diastolic (DBP) blood pressure will be recorded. Participants will be classified as hypertensive if SBP ≥ 140mm/Hg, and/or DBP ≥ 90mm/Hg [30]. If the automated blood pressure measurement repeatedly fails, for example when the reading is distorted by movement of the cuff, we will use a sphygmomanometer to manually measure the participants’ blood pressure.

#### 2.6.2. Obesity, Waist, and Hip Circumference

A trained researcher will use a measuring tape (Seca 200, Seca, Hamburg, Germany) to measure waist and hip circumference [32]. Waist circumference will be measured at the midpoint between the lowest palpable rib and the top of the iliac crest. Hip circumference will be measured around the widest portion of the buttocks. Each measurement will be repeated twice. If the repeated measurements are within 1cm of one another, we will calculate an average score (cm). If the difference between the measurements exceeds 1cm, the two measurements will be repeated [32]. Male participants will be classified as obese if their waist circumference >94 cm and their hip-waist ratio ≥0.90, and female participants if their waist circumference >80 cm and their hip-waist ratio ≥0.85 [32].

#### 2.6.3. Hypercholesterolemia, Diabetes, and Metabolic Syndrome

The medical staff of the care providers will collect blood samples after overnight fasting. The samples will be analysed at the laboratory of the Erasmus MC. We will measure total cholesterol (TC), low-density lipoprotein (LDL-C), high-density lipoprotein (HDL-C), triglycerides (TG), HbA1c and fasting glucose (FG) in mmol/L. Participants will be classified as having hypercholesterolemia when TC > 4 mmol/L and LDL-C ≥ 2.6 mmol/L [4]. Cut-of values for classifying participants as having diabetes mellitus type 2 are set at ≥48 mmol/L for HbA1c and ≥7 mmol/L for FG [31]. Participants will be classified as having metabolic syndrome when at least three of the following CVRF are present: elevated waist circumference (≥94 cm for men, ≥80 cm for women), elevated blood pressure (SBP ≥ 130 mm/Hg, and/or DBP ≥ 85 mm/Hg), elevated FG (>5.6 mmol/L), elevated triglycerides (>1.7 mmol/L), and low HDL-C (<1 mmol/L for men, < 1.3 mmol/L for women) [29].

### 2.7. Secondary Outcomes

#### 2.7.1. Physical Fitness, Muscle Endurance, Muscle Strength, Balance

A trained researcher will use the ID-fitscan to assess physical fitness. The ID-fitscan is a standardised set of performance tests that assesses muscle endurance (30 s chair stand and the five times chair stand), muscle strength (grip strength by hand-held dynamometry), and balance (four balance stances and comfortable walking speed). The set has been found feasible and reliable for adults with ID [36].

For the 30 s chair stand, participants are instructed to stand up and sit down as often as possible, without using their hands. We will record the number of complete stances performed in 30 s. For the five times chair stand, participants are instructed to stand up and sit down five times as fast as possible, without using their hands. We will record the time needed to complete five stances. For both tests, the participant starts sitting in the chair with the feet on the floor and the knees in a 90° angle.

We will use the Jamar dynamometer (Hydraulic, JAMAR, Sammons Preston Rolyan, Nottinghamshire, UK) to measure grip strength. Grip strength is tested in a seated position with the shoulder in 0° flexion and the elbow in 90° flexion. The participant is asked to squeeze the dynamometer with maximum force three times, with one minute rest in between. Both hands will be tested. The maximum produced force of the six attempts (kg) is the final test score.

Static balance will be measured with four stances that increase in difficulty: side-by-side stand, semi-tandem stand, tandem stand, and single leg stand. The participant must try to maintain each stand independently for 10 s for both sides, with a maximum of five attempts. If successful, the participant continues to the next stand. We will record the time for each completed stance and use the scores for side-by-side stand, semi-tandem stand, and tandem stand to categorise static balance from 0 (worse) to 4 (best) [37]. Dynamic balance will be measured as comfortable gait speed on an 11 m walkway. The participant will start from stance and accelerate to comfortable gait speed in the first 3 m of the walkway. When crossing the 3 m mark, a stopwatch is put in action and gait speed will be measured over the following 5 m. At the 8 m mark timing will be ended, and the participant will decelerate to standing still over the last 3 m of the walk. The test will be repeated three times. The average time of the three trials (seconds) will be converted to gait speed (m/s).

#### 2.7.2. Sarcopenia, Body Composition

We will use bioelectrical impedance analysis (Tanita Body Composition Analyzer MC-780, Tanita, Tokyo, Japan) to measure body mass (kg), fat mass (kg and %), and muscle mass (kg and %). The Tanita MC-780 estimates muscle mass based on whole body conductivity and uses a conversion equation that is calibrated with Dual-energy X-ray absorptiometry as a reference [38]. Muscle mass will also be measured by calf circumference. A trained researcher will use a measuring tape (Seca 200, Seca, Hamburg, Germany). Calf circumference will be measured twice at the widest point of the calf. If both measurements are within 1cm of one another, we will calculate an average score (cm). If the difference between the measurements exceeds 1cm, the two measurements will be repeated.

Table 3 presents the cut-off values that we will use to classify participants as being sarcopenic [17]. We will classify participants that score under the cut-off values for muscle endurance, muscle strength and muscle mass as sarcopenic. If participants also score under the cut-off value for comfortable gait speed (≤0.8 m/s), they will be classified as severe sarcopenic [17].

#### 2.7.3. Physical Activity

To assess the physical activity levels, we will use the International Physical Activity Questionnaire (IPAQ) [39]. The IPAQ assesses daily physical activity over the last seven days. The IPAQ consists of 27 items that are divided into five domains of physical activity: job-related, transportation, housework, leisure time, and sitting. For each domain, scores are provided for walking, moderate-intensity activities, and vigorous-intensity activities. Scores will be reported as the estimation of metabolic equivalent-minutes per week (MET-minutes/week) [40]. The IPAQ has been found reliable and valid in the general population [39].

#### 2.7.4. Activities of Daily Living

We will use the Barthel Index [41] to assess the ability to perform basic activities of daily living. The Barthel Index consists of ten items (bowel control, bladder control, grooming, toilet use, feeding, transfer, walking, dressing, climbing stairs, and bathing) with item scoring categories ranging from two to four categories. The total score ranges from 0 (completely dependent) to 20 (completely independent) [42]. The Barthel Index has good psychometric properties in the general population [42].

To assess the ability to perform instrumental activities of daily living (IADL), we will use an adapted version of the Lawton IADL scale [43]. We expect that adding and rephrasing some items for preparing food, housekeeping, and doing laundry, will increase the responsiveness of the scale. In a previous study we found this version to be applicable in older adults with ID [44]. The items were derived from the Groningen Activity Restriction Scale (GARS) [45]. All items will be scored on a 4-point rating scale ranging from 0 (not at all, with complete help) to 3 (fully independent, without any difficulty). All items scores will be summed. A detailed description of the items and scoring is presented in Table 4. We will administer the Barthel Index and the adapted version of the Lawton IADL scale as an informant-based questionnaire, completed by the professional caregiver of the participant.

#### 2.7.5. Falls

We defined a fall as an event that results in a person coming to rest inadvertently on the ground, floor, or other lower level [46]. We will use a fall calendar during the baseline period and the 12-week follow-up period. Falls are logged daily by the participant or the involved professional caregiver by putting a green sticker (no fall) or a red sticker (fall) on the calendar. If a participant falls multiple times during the day, the number of falls will be written down. The professional caregiver will check the calendar for completeness. For each period, we will classify participants as a non-faller or a faller (≥1 fall).

#### 2.7.6. Challenging Behaviour

We will evaluate challenging behaviour with the Aberrant Behaviour Checklist (ABC) [47]. The ABC consists of 58 items that are divided into five subscales: irritability, lethargy, stereotypic behaviour, hyperactivity, and inappropriate speech. Items are scored on a four-point scale ranging from 0 (not a problem) to 3 (severe problem) by the professional caregiver of the participant. The summed score of each subscale will be recorded separately [47]. The ABC is a reliable and valid instrument in adults with ID [48].

### 2.8. Adverse Events

Adverse events are defined as any undesirable experience occurring to a participant during the study. It does not matter if the event is related or unrelated to the experimental intervention. Adverse events can be reported spontaneously by the participant, trainer, or professional caregiver. All adverse events will be recorded by the researcher.

### 2.9. Participant Characteristics

Age, sex, ethnicity, level of ID, aetiology of ID, family history of CVRF (yes/no), kidney disease (yes/no), and medication use related to CVRF, are collected from medical records. The level of ID will be categorised as mild (IQ = 50–69) or moderate (IQ = 35–49). Professional caregivers will provide information about residential setting, sport activities, smoking, and alcohol use.

### 2.10. Procedures

Figure 1 shows a schematic representation of the study procedures. Participant characteristics will be collected at T0. During the 12-week baseline period, measurements for the primary and secondary outcomes will take place at six-week intervals (T0, T1, and T2). A minimum of three measurements is needed to establish a valid baseline and to filter out the natural variability that may exist in these parameters without applying any intervention [49]. We will record changes in sport activities and medication use related to CVRF. The falls will be tracked continuously throughout the 12-week baseline period.

After the baseline period, the intervention period starts for a duration of 24 weeks. We will record changes in sport activities and medication use related to CVRF. At the end of the intervention period all measurements for the primary and secondary outcomes will be repeated (T3).

The follow-up period will last 12 weeks. Directly post-intervention, we will ask all participants to fill out a questionnaire about their experiences with the PRET programme. We plan interviews with participants and their trainers who want to continue the PRET programme and evaluate facilitators and barriers for successful exercise in their daily life. At the end of the follow-up period all measurements for the primary and secondary outcomes will be repeated (T4). Falls will be tracked continuously throughout the follow-up period. Participants who drop out prematurely from the intervention, will be invited for the measurements at T3 and T4.

### 2.11. Statistical Analysis Plan

#### 2.11.1. Baseline Characteristics

Descriptive statistics will be used to present group characteristics for age, sex, ethnicity, level and aetiology of ID, family history of CVRF, kidney disease, use of medication, and primary and secondary outcomes at T0. We will use the mean and standard deviation (SD) to present normally distributed data, the median and interquartile range for non-normally distributed data, and the number and percentages for categorical variables.

#### 2.11.2. Effect of PRET Programme (Post-Intervention)

We will use two-level hierarchical linear and logistic regression models to investigate the effect of the PRET programme at the end of the intervention period (T3). The first level consists of the repeated measurements nested within a participant; the second level consists of the clustering of participants by the trainer delivering the PRET programme. We will use the likelihood ratio test (correcting for the boundary) to assess the necessity for random coefficients in the regression model. Statistical analyses will be performed with R-statistics (R Foundation for Statistical Computing, Vienna, Austria) and we will use the package Linear and Nonlinear Mixed Effects Models (nmle) [50]. A two-tailed significance level of 0.05 will be used for all tests.

The variables for the PRET programme will be coded as 0 (baseline period at T0, T1, and T2) or 1 (intervention period T3). To check the assumption of stable baseline measurements, we will investigate the effect of time on each outcome variable assessed during the baseline period. Time varying variables (changes in sport activities and medication use related to CVRF) will be included as covariates in all models.

We will use the data from all participants who started the study. We will also analyse the effect of the intervention for participants who completed the intervention as planned.

#### 2.11.3. Follow-Up of Outcomes (12 Weeks Post-Intervention)

We will use two-level hierarchical linear and logistic regression models to investigate the change in primary and secondary outcomes 12 weeks after ending the PRET programme (T3-T4). Time varying variables (medication use and physical activity) will be included as covariates. We will explore if results differ based on group characteristics (age, sex, level and aetiology of ID) by adding them subsequently to the linear and logistic regression models.

#### 2.11.4. Falls during Baseline and 12-Week Follow-Up

We will use the McNemar test to compare the numbers of non-fallers and fallers during the baseline period and the 12-week follow-up period.

### 2.12. Sample Size Calculation

We calculated the sample size for the outcome HbA1c using simulated data and repeated the simulation a 1000-times in a hierarchical way. To determine the effect of the trainers we used the outcome of our pilot study [25] where we found a variation of 30% for the clustering of participants by the trainer who delivered the PRET programme. The type 1 error (alpha) was set at 0.05 and power was set at 80%. We used an effect size (mean change in %HbA1c = −0.55, SD = 0.47) derived from a study that examined the effect of high intensity resistance training in adults with diabetes mellitus type 2 [10]. This resulted in a sample of 35 participants needed for this study. In our pilot study, 58% of the participants completed the PRET programme successfully at intended intensity [25]. Therefore, we inflate the number of participants by 42% and aim to enrol 61 participants in this study.

## 3. Discussion

In this study we will examine the effects of the PRET programme on CVRF, physical fitness, sarcopenia, activities of daily living, falls, and challenging behaviour in adults with mild to moderate ID. These health issues are highly prevalent in this population [3,15,21,22,23] and non-pharmacological treatment strategies are needed to reduce the impact of these problems on health. If the PRET programme is effective in reducing CVRF, then it may be added as a strategy to existing interventions to reduce cardiovascular risk in adults with ID. Additionally, gaining muscular strength may increase physical fitness levels, reduce sarcopenia, enhance the ability to perform activities in daily life and reduce falls. The programme may have a positive impact on challenging behaviour.

A randomised controlled trial is often considered the gold standard for intervention research and offers the ability to reduce bias due to confounding [51]. We considered a randomised study design; however, randomised controlled trials in people with ID have been challenging. Obtaining informed consent from legal representatives, rather than from eligible participants themselves, often results in small sample sizes and insufficient statistical power [51]. We expect that legal representatives may refuse study participation, as the participant they represent may be allocated to the control group and will not receive the PRET programme. Therefore, we opted for a repeated time series design with one study group. Because of the single group design, we can offer all participants the PRET programme, and all participants may profit from possible health benefits. The use of the repeated measures within each participant will increase the statistical power, and the multilevel regression analyses allow us to adjust for time varying variables such as sport activities and medication use related to CVRF.

Our study design has some limitations. First, we are not able to control for elements of care that we will not measure. This may result in uncontrolled residual confounding. Second, although the measurements used to assess the primary outcomes are objective, the validity of the measurements to assess physical fitness may be influenced by the ability of the participants to follow instructions. However, the results of the physical fitness tests will only be recorded if the trained researcher is convinced that participants understand how to perform the test. Finally, the lack of blinding of the researcher performing the assessments may introduce measurement bias.

A strength of this study is the clinically applicability of the PRET programme. In a pilot study [25] we created a set of feasible resistance exercises for adults with ID (RESID) that we used in the PRET programme. Secondly, we assessed the feasibility of the PRET programme in adults with ID and CVRF [26]. We found that the PRET programme was feasible when close supervision was provided to the participant, with 58% of the participants achieving vigorous training intensity.

Adherence to the PRET programme is important for the internal validity of the study. It may be challenging for adults with ID to complete a 24-week exercise programme because of motivational factors. Based on our experiences from the pilot and feasibility study, we will incorporate several strategies (a progression certificate to track progress, a T-shirt to wear during the training and measurements, and the participant receives a certificate and a medal at the end of the programme) to maximize attendance and motivation of the participants. In addition, the training will be provided by trainers who are experienced in motivating adults with ID when performing physical exercises.

## 4. Conclusions

To our knowledge, there is no study that examined the effect of resistance training on CVRF in adults with ID. If the PRET programme can reduce CVRF, this intervention may have major implications for clinical practice. In this case, the PRET programme could be used as an additional intervention to the existing pharmacological management of CVRF in adults with ID.

## Figures and Tables

**Figure 1 ijerph-19-16438-f001:**
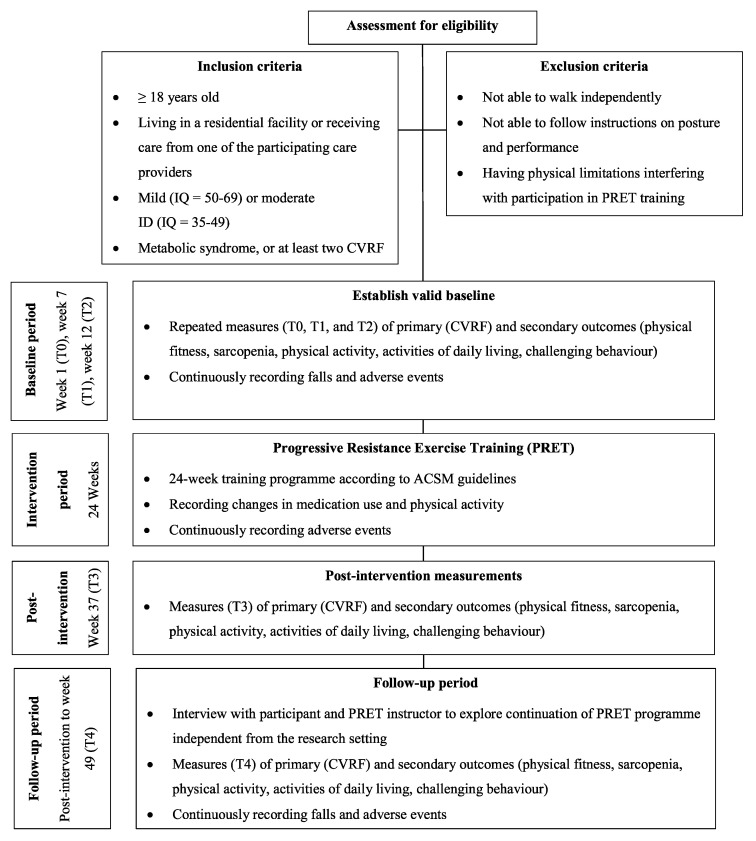
Study recruitment and procedures. ACSM: American College of Sports Medicine; CVRF: cardiovascular risk factors; ID: intellectual disability IQ: intelligence quotient.

**Table 1 ijerph-19-16438-t001:** Definitions of cardiovascular risk factors (CVRF).

Cardiovascular Risk Factor	Reference	Markers	Values
Metabolic syndrome, having 3 out of 5 markers	Alberti et al. (2009) [29]	Waist circumference	≥94 cm (men) * ≥80 cm (women) *
		Fasting glucose (FG)	>5.6 mmol/L
		High density lipoprotein (HDL-C)	<1 mmol/L (men) <1.3 mmol/L (women)
		Triglycerides (TG)	>1.7 mmol/L
		Systolic blood pressure (SBP) and diastolic blood pressure (DBP)	≥130 mm/Hg and ≥85 mm/Hg
Hypertension (Stage 2)	Whelton et al. (2017) [30]	Systolic blood pressure (SBP)	≥140 mm/Hg
		Diastolic blood pressure (DBP)	≥90 mm/Hg
Hypercholesterolemia	Piepoli et al. (2016) [4]	Total cholesterol (TC)	>4 mmol/L
		Low density lipoprotein (LDL-C)	≥2.6 mmol/L
Diabetes mellitus type 2	American Diabetes Association (2018) [31]	Fasting glucose (FG)	≥7 mmol/L
		HbA1c	≥48 mmol/L
Obesity	World Health Organisation (2008) [32]	Waist Circumference	>94 cm (men) * >80 cm (women) *
		Waist-hip ratio	≥0.90 (men) ≥0.85 (women)

* European, Caucasian threshold, mm/Hg: millimetres of mercury; mmol/L: millimoles per litre.

**Table 2 ijerph-19-16438-t002:** Training parameters according to the ACSM guidelines [11].

Mesocycle	% of 1RM	Number of Sets	Number of Repetitions	Rest between Sets
1	-	1–2	20	30 s
2	65	2	15	1 min
3	75	3	10	1 min
4	80	3	8	2 min

ACSM: American College of Sports Medicine; 1RM: one-repetition maximum.

**Table 3 ijerph-19-16438-t003:** Classification for Sarcopenia cut-off [17].

Domain	Tests	Values
*Sarcopenia*		
Muscle strength	Five times chair stand	>15 s for 5 rises
	Grip strength	<27 kg (men) <16 kg (women)
Muscle mass	BIA	<20 kg (men) <15 kg (women)
*Severe sarcopenia*		
Muscle strength	Five times chair stand	>15 s for 5 rises
	Grip strength	<27 kg (men) <16 kg (women)
Muscle mass	BIA	<20 kg (men) <15 kg (women)
Physical performance	Gait speed	≤0.8 m/s

BIA: bioelectrical impedance analysis; kg: kilogram; m/s: meter per second.

**Table 4 ijerph-19-16438-t004:** Items and adapted scoring based on Lawton and Brody IADL.

Items Lawton and Brody IADL [43]	Items derived from GARS [45]	Items and Adapted Scoring Used in This Study
*Items*		
Using the telephone		Using the telephone
Shopping		Shopping
Preparing food	Prepare breakfast or lunch	Prepare breakfast or lunch
	Prepare diner	Prepare dinner
Housekeeping	Light household activities	Light household activities
	Heavy household activities	Heavy household activities
	Make the bed	Make the bed
Doing laundry	Wash and iron clothes	Wash and iron clothes
Using transportation		Using transportation
Handling medication		Handling medications
Handling finances		Handling finances
*Scoring of items*		
0 = Less able	5 = Not at all, with complete help	0 = Not at all, with complete help
1 = More Able	4 = Not fully independent, but with some help	1 = Not fully independent, but with some help
	3 = Fully independent but with great difficulty	2 = Fully independent but with great difficulty
	2 = Fully independent but with some difficulty	3 = Fully independent without any difficulty
	1 = Fully independent without any difficulty	

GARS: Groningen Activity Restriction Scale; IADL: instrumental activities of daily living.

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
