# Peer review of "The Effect of Progressive Resistance Exercise Training on Cardiovascular Risk Factors in People with Intellectual Disabilities: A Study Protocol"

_ijerph, 2022, doi:10.3390/ijerph192416438_

Round 1

Reviewer 1 Report

The manuscript “The effect of progressive resistance exercise training on cardiovascular risk factors in people with intellectual disabilities: a study protocol” aimed to propose a study protocol to investigate the impacts of resistance training to reduce cardiovascular risk in people with intellectual disabilities. I commend the author for their excellent work in preparing the manuscript. General comments and specific points and sections are provided below:

General comments

Presentation: writing is good and figures/tables are clear and informative

Title: the title is clear and informative.

Abstract: the writing is clear and concise, and the abstract provides all the necessary information

Introduction: the introduction is well written, and successfully provides the information for the rationale of the study.

Materials and Method: the material and methods section is also well-written and provides detailed information regarding the protocol of the study. The authors also provide a figure resuming the stages of the experiment. All methodological aspects of the study are thoroughly detailed, and I commend the authors for their excellent work.

Discussion: the discussion is brief (as should be for this type of manuscript) and provides important interpretations. It successfully discusses the sole limitation of the study, which is the adoption of a quasi cross-over design instead of a randomized controlled trial. This reviewer agrees with the purposes for this choice, but recommends that the authors also list the limitations associated with the design they chose.

Author Response

We would like to thank you for your critical review of our manuscript. We are pleased to hear that overall, the manuscript was clear and well-written. In the discussion we added the limitations for the chosen design: Page 11, lines 388-390

‘A limitation of this study design is that we are not able to control for elements of care that we will not measure, which may result in uncontrolled residual confounding.’  

Hopefully we have addressed the comment sufficiently. 

Reviewer 2 Report

The protocol is well designed and written.

It will be original work.

Control is baseline values. Wouldn't it be more robust to make a control group without intervention and/or with aerobic exercise?

The authors follow the 2014 ACSM guidelines (reference 11). However, the 2022 ACSM guidelines already exist, which should be followed, although there is no significant difference from 2014.

There are several subgroups in people with intellectual disabilities – shouldn't there be a subgroup analysis?

We await answers to these questions.

Author Response

We would like to thank you for your critical review of our manuscript. We are pleased to hear that the protocol is well designed and written. We provide our responses to the comments below in a point-by-point fashion. 

  1. The protocol is well designed and written. It will be original work. Control is baseline values. Wouldn't it be more robust to make a control group without intervention and/or with aerobic exercise?

It would certainly be more robust to use a controlled design with a control group without the intervention. We considered a randomised controlled study design; however, randomised controlled trials in people with ID have been challenging. Obtaining informed consent from legal representatives, rather than from eligible participants themselves, often results in small sample sizes and insufficient statistical power. Thereby, because of the single group design, we can offer all participants the PRET programme, and all participants may profit from possible health benefits. This choice for our design is described in the discussion section as well (see page 11, lines 379-392). 

  1. The authors follow the 2014 ACSM guidelines (reference 11). However, the 2022 ACSM guidelines already exist, which should be followed, although there is no significant difference from 2014.

Thank you for alerting us on the ACSM reference. We have verified and edited the reference to the most recent ACSM guidelines. 

  1. There are several subgroups in people with intellectual disabilities – shouldn't there be a subgroup analysis? We await answers to these questions.

For the analysis we are indeed planning to analyse different subgroup. Thank you for pointing out that this is not clearly described in the manuscript. We added the following:  ‘We will explore if results differ based on group characteristics (age, sex, level and aetiology of ID) by adding them subsequently to the linear and logistic regression models.’ 

Page 11, lines 351-353. 

Hopefully we have addressed the comments sufficiently. 

Reviewer 3 Report

These authors are proposing a study to verify cardiovascular risk factors of people with intellectual disability after a protocol of progressive resistance exercise training of 24 weeks. The proposal was well written, detailed and designed.

The group have previous experience and published studies attesting the application and feasibility of the protocol in the target population. If rightly conduced, the progressive resistance exercise training protocol seems to be feasible to be executed by people with intellectual disability.

The introduction has valid arguments and the progressive resistance exercise training has been shown to reduce cardiovascular risk factors on the general population. There is some reason why the progressive resistance exercise training would not have the same outcomes on people with intellectual disability? If yes, please clarify it on the introduction.

The proposal is a longitudinal design with repeated measures and one group to avoid that the legal representatives refuses to participate as the participant they represent may be allocated on the control group. This design would result in a higher statistical power.

In the methods sections, the authors did a very good job in idealizing the experimental design and describing the procedures for a good conduction of the study. The authors have adopted good practices to individualize training and exercise intensity prescription by using the 1RM parameter and load monitoring methods. The assessments of the primary and secondary outcomes are totally based on valid references and even adverse events were previewed.

The study inclusion and exclusion criteria is valid. However, the training adherence may vary largely due to the long duration of the protocol. I would like to suggest the inclusion of an exclusion criteria based on training adherence (e.g. minimum adherence value). Lastly, I would like to suggest to be included some valid limitations and hypothesis for the study outcomes.

Author Response

We would like to thank you for your critical review of our manuscript. We are pleased to hear that you found our manuscript well written, detailed and designed. We provide our responses to the comments below in a point-by-point fashion. 

  1. The introduction has valid arguments and the progressive resistance exercise training has been shown to reduce cardiovascular risk factors on the general population. There is some reason why the progressive resistance exercise training would not have the same outcomes on people with intellectual disability? If yes, please clarify it on the introduction.

Thank you for this critical remark. We added a section in the introduction about the differences between people with ID and the general population. Page 1, lines 45-51. 

‘Studies indicate that the physiological responses to exercise are different in people with ID. For example, there are indicators that they have a blunted blood pressure and baroreflex sensitivity response and a lower maximal heart rate and peak oxygen uptake than people in the general population. Therefore, the effect of exercise interventions may differ for this population, and we can’t generalize the results from the general population to people with ID.’   

  1. The study inclusion and exclusion criteria is valid. However, the training adherence may vary largely due to the long duration of the protocol. I would like to suggest the inclusion of an exclusion criteria based on training adherence (e.g. minimum adherence value).

Thank you for this comment. The training adherence could indeed vary largely in this population. The potential participants have been screened by a physical therapist for their eligibility and expected adherence to the programme. We hope that this will limit the variety on training adherence. We added this to the manuscript on page 3, lines 106-108

We will also take this into account in our statistical analyses. We will perform an intention-to-treat analysis and a per-protocol analysis (excluding participants that did not adhere to the training protocol) to investigate if non-compliancy has impact on our results. These analyses are described on page 11, lines 345-346

  1. Lastly, I would like to suggest to be included some valid limitations and hypothesis for the study outcomes.

 We are not sure if we understand this comment correctly. Hopefully we have interpreted your suggestion correctly, if not, please let us know.  

In the discussion of the manuscript we added the following: Page 12, lines 392-398. 

‘Our study design has some limitations. First, we are not able to control for elements of care that we will not measure. This may result in uncontrolled residual confounding. Second, although the measurements used to assess the primary outcomes are objective, the validity of the measurements to assess physical fitness may be influenced by the ability of the participants to follow instructions. However, the results of the physical fitness tests will only be recorded if the trained researcher is convinced that participant understands how to perform the test. Finally, the lack of blinding of the researcher performing the assessments may introduce measurement bias.’ 

Hopefully we have addressed the comments sufficiently.